# Accuracy of Real Time Continuous Glucose Monitoring during Different Liquid Solution Challenges in Healthy Adults: A Randomized Controlled Cross-Over Trial

**DOI:** 10.3390/s22093104

**Published:** 2022-04-19

**Authors:** Janis R. Schierbauer, Svenja Günther, Sandra Haupt, Rebecca T. Zimmer, Beate E. M. Zunner, Paul Zimmermann, Nadine B. Wachsmuth, Max L. Eckstein, Felix Aberer, Harald Sourij, Othmar Moser

**Affiliations:** 1Division of Exercise Physiology and Metabolism, Department of Sport Science, University of Bayreuth, 95440 Bayreuth, Germany; janis.schierbauer@uni-bayreuth.de (J.R.S.); svenja.guenther@uni-bayreuth.de (S.G.); sandra.haupt@uni-bayreuth.de (S.H.); rebecca.zimmer@uni-bayreuth.de (R.T.Z.); beate.zunner@uni-bayreuth.de (B.E.M.Z.); paul.zimmermann@uni-bayreuth.de (P.Z.); nadine.wachsmuth@uni-bayreuth.de (N.B.W.); max.eckstein@uni-bayreuth.de (M.L.E.); felix.aberer@uni-bayreuth.de (F.A.); 2Interdisciplinary Metabolic Medicine Trials Unit, Division of Endocrinology and Diabetology, Department of Internal Medicine, Medical University of Graz, 8036 Graz, Austria; ha.sourij@medunigraz.at

**Keywords:** continuous glucose monitoring, sensor accuracy, sodium chloride, glucose, Ringer’s solution, healthy individuals, liquid consumption

## Abstract

Continuous glucose monitoring (CGM) represents an integral of modern diabetes management, however, there is still a lack of sensor performance data when rapidly consuming different liquids and thus changing total body water. 18 healthy adults (ten females, age: 23.1 ± 1.8 years, BMI 22.2 ± 2.1 kg·m^−2^) performed four trial visits consisting of oral ingestion (12 mL per kg body mass) of either a 0.9% sodium chloride, 5% glucose or Ringer’s solution and a control visit, in which no liquid was administered (control). Sensor glucose levels (Dexcom G6, Dexcom Inc., San Diego, CA, USA) were obtained at rest and in 10-min intervals for a period of 120 min after solution consumption and compared against reference capillary blood glucose measurements. The overall MedARD [IQR] was 7.1% [3.3–10.8]; during control 5.9% [2.7–10.8], sodium chloride 5.0% [2.7–10.2], 5% glucose 11.0% [5.3–21.6] and Ringer’s 7.5% [3.1–13.2] (*p* < 0.0001). The overall bias [95% LoA] was 4.3 mg·dL^−1^ [−19 to 28]; during control 3.9 mg·dL^−1^ [−11 to 18], sodium chloride 4.8 mg·dL^−1^ [−9 to 19], 5% glucose 3.6 mg·dL^−1^ [−33 to 41] and Ringer’s solution 4.9 mg·dL^−1^ [−13 to 23]. The Dexcom G6 CGM system detects glucose with very good accuracy during liquid solution challenges in normoglycemic individuals, however, our data suggest that in people without diabetes, sensor performance is influenced by different solutions.

## 1. Introduction

Continuous glucose monitoring (CGM) became a cornerstone of modern diabetes management [1]. Although long-term glycemic management is still assessed by means of glycated hemoglobin (HbA_1c_) levels, CGM systems offer, next to the mean interstitial glucose level and Glucose Management Indicator (GMI), the opportunity to assess, the time spent below and above the physiological glucose range (79–101 mg·dL^−1^) for healthy individuals as well as glucose variability [2]. The accordance between the GMI and the laboratory assessed HbA_1c_ levels have been shown to be valid, detailing the one or the other might be used together with additional sensor glucose parameters [3]. However, the accordance between GMI and HbA_1c_ levels might also be depending on the CGM sensor accuracy. Hence, numerous studies have investigated the CGM performance around different glycemic challenges, including oral glucose tolerance testing [4], physical activity and exercise [5], as well as during the nocturnal period [6]. In addition to the importance of receiving accurate CGM metrics for the retrospective assessment of therapy and diabetes management, especially people with type 1 diabetes and their health care professionals need to rely on accurate sensor glucose levels. Over- and/or underestimation of actual blood glucose levels by the CGM sensor might result in inappropriate therapy decisions, thus increasing the risk of severe hypo- or hyperglycemia. Even though research has shown that episodes of a high rate of change in glucose [5], hypoglycemia [6] and partly exercise [7] remain the *Achilles Heel* of current CGM systems, the consumption of different liquid solutions has, to our knowledge, not yet been investigated. However, this knowledge would be of particular importance since the rapid intake of solutions might change the extra- and/or intracellular volumes and thus impair sensor accuracy. The importance of investigating different solutions on CGM performance is of great significance: in the clinical management sodium chloride and Ringer’s solutions are widely used for balancing hydration. Taking the need for accurate sensor glucose levels into account, the aim of this study was to evaluate real-time CGM (rtCGM) sensor performance during different oral solution challenges in healthy adults as a proof-of-concept study.

## 2. Materials and Methods

This was a single-center, randomized, controlled crossover trial, assessing the impact of different liquids on rtCGM sensor accuracy (Dexcom G6, Dexcom Inc., San Diego, CA, USA) tested against reference blood glucose levels in healthy individuals. The local ethics committee of the University of Bayreuth (Germany) approved the study protocol (O 1305/1-GB, 10 June 2021). The study was conducted in conformity with the declaration of Helsinki and guidelines for good clinical practice. Before any trial-related activities, potential participants were informed about the study protocol and participants gave their written informed consent.

### 2.1. Eligibility Criteria

Eligibility criteria included male or female individuals aged between 18–65 years with a body mass index (BMI) of 18.0–29.9 kg·m^−2^, both inclusive. Individuals were excluded if they were enrolled in a different study, received any kind of medicinal product including investigational medicinal products, and had a blood pressure outside the range of 90–150 mmHg for systolic and 50–95 mmHg for diastolic after resting for five minutes in a supine position. Furthermore, participants were excluded if they suffered from any kind of metabolic disease, including renal or thyroid, or had a history of multiple and/or severe allergies to any trial-related products. To assure an euhydrated status prior to the study experiments, participants were also excluded if they demonstrated a urine-specific gravity outside the range of 1005–1030 mg·mL^−1^.

### 2.2. Assessment of Eligibility

Inclusion and exclusion criteria were assessed by an investigator at the screening visit prior to the start of the study.

### 2.3. Study Design

After inclusion in the study, participants were assigned to ascending numbers and then allocated to the order in which the trial visits were conducted following a cross-over randomized fashion with the software Research Randomizer^®^ (1:1:1:1) [8]. Prior to the first visit, participants received the rtCGM sensor, which was worn on the lateral upper arm and calibrated immediately before each trial visit. Participants received an individual quantity of each solution calculated on the basis of body mass, which was measured at each trial visit. Between each trial visit, a minimum period of 48 h was maintained, except for the control condition, after which a further visit took place after a minimum of 24 h.

### 2.4. Trial Visits

Prior to the start of each trial visit, participants had to fast for at least 12 h and refrain from any strenuous physical activity for at least 48 h. Participants were also not allowed to consume any alcohol within 24 h before the visit. Sensor insertion was performed 24 h before the first trial visit. When attending the research facility in the morning after an overnight fast, a urine-specific gravity test was performed (Combur^10^, Roche Deutschland Holding GmbH, Grenzach-Whylen, Germany). Subsequently, the participants remained in a seated position for 10 min before a duplicate baseline analysis of body composition was conducted using a bioelectrical impedance analysis (Inbody 720, Inbody Co., Seoul, Korea). Afterwards, they received 12 mL per kg body mass of a sodium chloride (NaCl 0.9%, B. Braun, B. Braun Melsungen AG, Melsungen, Germany), glucose (Glucose-Lösung 5%, Deltamedica, Reutlingen, Germany) or Ringer’s solution (Ringer B. Braun, Melsungen AG, Melsungen, Germany), respectively, which had to be consumed within 90 s. Interstitial glucose via continuous glucose monitoring (Dexcom G6, Dexcom Inc., San Diego, CA, USA) and reference glucose values from capillary blood from the earlobe (Biosen S-line, EKF Diagnostics, Barleben, Germany) were obtained at rest, immediately after the fluid consumption and then every 10 min for 120 min. Any form of food and liquid intake was prohibited during each of the trial visits. Participants remained in a seated position at standardized room temperature and humidity for the entire time of the visit.

### 2.5. Statistical Analysis

All data were assessed for normal distribution by means of the Shapiro-Wilk normality test. Descriptive statistics are given as mean ± standard deviation (SD). Blood glucose was compared against sensor glucose for the same time point by median absolute relative difference (MedARD) analysis for overall data and the Bland-Altman method (bias and 95% limits of agreement). A comparison of absolute relative differences (ARD) between rtCGM and reference blood glucose was performed with the Friedman test with Dunn’s multiple comparisons. Statistical analyses were performed with Microsoft Excel (Microsoft Corporation 2007, Redmond, WA, USA) and the Prism Software version 8.0 (GraphPad, La Jolla, CA, USA).

## 3. Results

In total, 18 healthy individuals (ten females) were included in the study with a mean ± SD age of 23.1 ± 1.8 years, a height of 176 ± 10 cm, body mass of 69.5 ± 12.5 kg and BMI of 22.2 ± 2.1 kg·m^−2^. Skeletal muscle mass was 31.9 ± 7.1 kg, fat mass was 13.0 ± 3.8 kg (19.0 ± 5.3%) and total body water was 41.3 ± 8.6 L. All screened participants were eligible to participate in the study in which no participant had to be discontinued or left the study prematurely.

### 3.1. Median Absolute Relative Difference (MedARD)

Data were not normally distributed. The MedARD for the different solution challenges and the control condition are given in Table 1. When comparing the absolute relative differences (ARD), the 5% glucose challenge was significantly different from the control condition (*p* < 0.0001), sodium chloride (*p* < 0.0001) and the Ringer’s solution (*p* = 0.001). No differences were found between the other challenges.

### 3.2. Bland-Altman Analysis

The Bland-Altman method derived bias ± SD of bias and levels of agreement for absolute values of glucose for overall, sodium chloride, 5% glucose, Ringer’s solution and the control condition (rtCGM to BG) were found to be 4.3 ± 12.1 (−19.6, 27.9) mg·dL^−1^, 4.8 ± 7.1 (−9.1, 18.8) mg·dL^−1^, 3.6 ± 18.9 (−33.5, 40.8) mg·dL^−1^, 4.9 ± 9.1 (−12.9, 22.6) mg·dL^−1^ and 3.9 ± 7.4 (−10.5, 18.4) mg·dL^−1^, respectively (Figure 1a–f).

## 4. Discussion

In our study, we found that the rtCGM (Dexcom G6, Dexcom Inc., San Diego, CA, USA) measured interstitial glucose levels with very good accuracy when compared to reference blood glucose detailing a MedARD of 7.1% (*n* = 803). This good performance during different solution challenges is confirmed under pre-intervention conditions, in which a MedARD as low as 2.6% was found. Intriguingly, as soon as a liquid glucose solution was consumed, sensor accuracy slightly deteriorated when compared with the other solutions or control arm (MedARD 11.0%), which might be also related to the rate of change in glucose [9,10].

In a different study performed on pediatric individuals undergoing total pancreatomy with islet autotransplantation, a MedARD of 13.3% was found [11], detailing a reduced sensor performance as seen in our study. Further studies were performed in people with type 1 diabetes under different conditions, showing higher MedARDs as seen in our study that might be mainly explained by exaggerated glucose fluctuations as known in this metabolic disease [12,13,14,15,16]. From a physiological point of view, a certain lag time can be observed for the glucose to diffuse from the bloodstream into the interstitial space [4]. Most likely, a combination of physiologic perturbations and the way the algorithm considers recent signals, on top of the inherent inaccuracy, are responsible for the deterioration in sensor performance while consuming glucose. When considering the Ringer and sodium chloride solutions used in this study, it is worth mentioning that the theoretical osmolarity of these solutions was 308 and 309 mOsm·L^−1^, respectively, which is only slightly higher than the osmolarity of normal serum [17]. However, since even small changes in osmolarity can be expected to have a marked effect on water transfer [17,18], it is possible that sensor performance might also be impaired by these changes, although the absolute fluid intake in this study was rather small [19].

As CGM devices are now more widely available [20], e.g., for healthy people and athletes, their performance and sensor accuracy need to be further investigated to ensure accurate evaluation of glucose metabolism. Wrong conclusions based on over- and/or underestimation of actual blood glucose values might yield wrong assumptions of therapy adjustments in the general population (e.g., (pre)-diabetes, type 2 diabetes). Furthermore, inaccurate sensor glucose responses to different exercise sessions and meals might give coaches and nutritionists a misinterpreted evaluation of athletes’ performance. Especially, as different nutritional interventions are becoming more popular and their efficacy adherence are often assessed by means of CGM [21], inaccurate sensor data might also result in wrong clinical trial conclusions. Minimal sensor data differences in comparison of interventions (e.g., ~4–5 mg·dL^−1^) must be evaluated and discussed with caution, as these results might be based on interstitial glucose vs. blood glucose inaccuracy, as seen in our data.

When our data were assessed by means of the Bland-Altman method, in general sensor data tended to slightly overestimate actual reference blood glucose values. From a clinical point of view, absolute differences of around 4 mg·dL^−1^ play an insignificant role in yielding wrong therapeutic decisions, e.g., insulin injections in type 1 diabetes. Furthermore, our reference blood glucose values were evaluated by a laboratory tool, which also might have had minimal potency to inaccurately detect actual blood glucose values given an average deviation from the reference gold standard of 3.0% without any concentration-dependent variability [22].

Lastly, our study is not without limitations. First, the small number of participants resulted in a low number of points of comparison between sensor glucose values and reference blood glucose (~200). Furthermore, the solutions used in our study were comparable in terms of composition, except for glucose, which likely explains the slightly higher MedARD for the glucose solution, as in contrast to the other solutions, blood and interstitial glucose values changed. However, from our point of view, future studies need to assess the effects of different liquids and osmolarities on sensor performance in particular when infused intravenously as this will be important for different groups of hospitalized patients.

## 5. Conclusions

To the best of our knowledge, this is the first study performed for the Dexcom G6 rtCGM (Dexcom Inc., San Diego, CA, USA) assessing sensor glucose performance after consuming different liquid solutions in adults with normal glucose metabolism. Our data clearly showed that this rtCGM is accurately detecting actual blood glucose levels but tends to become inaccurate in response to oral fluid administration, detailing the highest MedARD when administering a glucose solution. In general, rtCGM minimally overestimated actual reference blood glucose levels (~4 mg·dL^−1^) which might be a random rather than a systematic discrepancy. Furthermore, based on our data and previous studies, it might be appropriate to postulate that sensor performance is accurate after the intake of a given amount of Ringer and sodium chloride solutions and that for the 5% glucose, some physiological (rather than technological) delays can be observed, mainly due to the diffusion of the glucose from the bloodstream into the interstitial space.

## Figures and Tables

**Figure 1 sensors-22-03104-f001:**
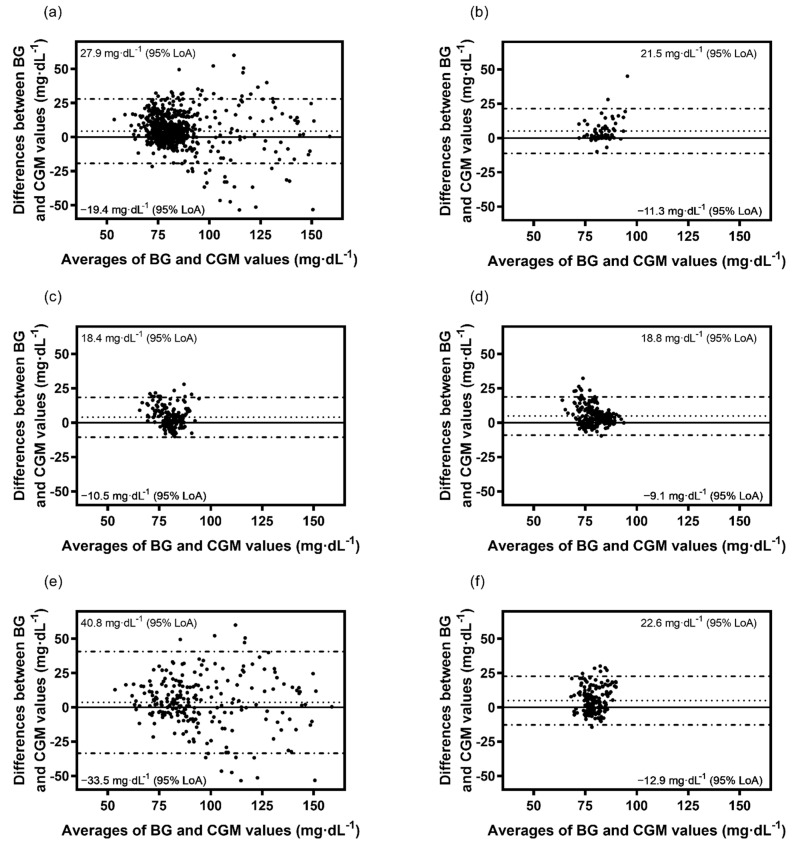
Bland-Altman plots for the comparison of BG and rtCGM values for the control visit and the three standardized liquids given as bias (dotted line) and levels of agreement (dot-dashed line): (**a**) Overall, (**b**) Overall baseline, (**c**) Control, (**d**) Sodium chloride, (**e**) 5% Glucose, (**f**) Ringer’s solution.

**Table 1 sensors-22-03104-t001:** Median absolute relative difference (MedARD) and interquartile range (IQR) between interstitial glucose and reference blood glucose for the four trial visits.

	rtCGM System Accuracy, Median Absolute Relative Difference * [IQR], %	*n*
Overall	7.1 [3.3–10.8]	803
Overall Baseline	2.6 [0.8–9.3]	186
Control	5.9 [2.7–10.8]	186
Sodium chloride	5.0 [2.7–10.2]	204
5% Glucose	11.0 [5.3–21.6]	227
Ringer’s solution	7.5 [3.1–13.2]	186

* Median absolute relative difference is expressed as a percentage. Median absolute relative difference indicates absolute values of difference and thus ignores the direction of the measurement error, but indicates the size of the error expressed as percentage error; *n* = number of points of comparison.

## Data Availability

Data will be made available upon reasonable request by the corresponding author.

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
