# Peer review of "Accuracy of Real Time Continuous Glucose Monitoring during Different Liquid Solution Challenges in Healthy Adults: A Randomized Controlled Cross-Over Trial"

_sensors, 2022, doi:10.3390/s22093104_

Round 1

Reviewer 1 Report

The authors evaluated the accuracy for glucose results obtained with a commercial CGM sensor (DEXCOM) vs Biosen S-line, EKF Diagnostics, from healthy adults under hydration with different oral solutions. The authors present the results from 18 participants, evaluating the CGM performance when the patient intake 4 different liquid solutions. Seems that experiments are well performed, however, the data analysis, discussion and, the conclusion can be improved.

The authors conclude that the CGM is accurate in detecting actual blood glucose levels, DEXCOM CGM is a dispositive already approved in the USA by the FDA, the conclusion is something already known. Besides, the authors said that the CGM results tend to become inaccurate when oral fluids are administrated, especially when the administrated solution contains glucose. Are the inaccurate results related to a delay between capillary blood and interstitial fluid glucose concentration? Have the authors considered comparing the results with capillary blood from the finger or blood from the vein? Could the author use the glucose golden standard for the comparison of their results?

For the statistical analysis, the authors mention they apply Shapiro-Wilk normal test, could the authors include the results from the Normality test. Did the authors evaluate normality for the complete sample or by groups of data?

The authors mentioned that results were obtained from samples of 18 healthy individuals, 10 women, how do the results from women perform compare to the results from men?

Have the authors considered that the inaccuracy of their results when the intake solution contains glucose could be due to a glucose tolerance of each individual?

Based on the results and observations, the discussion and conclusion need to be improved.

The reviewer recommends minor revision before considering the manuscript for its publication in Sensors.

Sincerely,

The reviewer

Reviewer 2 Report

The communication entitled “Accuracy of Real Time Continuous Glucose Monitoring during Different Liquid Solution Challenges in Healthy Adults: A Randomized Controlled Cross-Over Trial” provides sensor performance data of different oral solution on CGM in healthy adults. The results show that rtCGM is accurately detecting actual blood glucose levels but tends to overestimate actual reference blood glucose levels. The findings are interesting and the article is well structured, all the sections provide sufficient material for the clear understanding of the topic and results. I consider the manuscript adequate for its publication with minor changes, including general correction of the English spelling and some typing error along the text.

Author Response

Dear Reviewer,

first of all, we would like to thank you very much for your time and effort in reviewing our manuscript submission. We are grateful for your comments and suggestions and have re-checked the manuscript for spelling and typing errors, hoping that the language quality is now sufficient for publication. Lastly, we very much hope that our work can make a valuable contribution in this scientific area.

Yours sincerely,

Janis Schierbauer and co-authors 

Reviewer 3 Report

The authors have compared the performance of glucose monitoring sensors in healthy individuals post-consumption of various liquids. The study is designed systematically however the very less number of participants did not reflect the actual accuracy of the system. Still, there are a few changes that must be made to the manuscript before accepting for publication:

  1. Authors should consider discussing diagnostic accuracy by modeling the dependence of sensitivity and specificity.
  2. The biggest drawback of this study is the sample size which is rather very small (n=18) however, it would be good to discuss the difference in percent agreement between males and females in various tests.
  3. How was the 95% limit of agreement (LOA) calculated? The authors need to define the sample mean of the differences and sample standard deviation which was used to calculate LOA.
  4. The most important limitation of CGM is the interfering substances leading to false results. However, the current study did not discuss it.
  5. It is not clear what is the purpose of testing healthy individuals. It is well known that the range and fluctuations of glucose levels are much different in healthy and diabetic individuals which can affect the sensor performance. The authors need to discuss this in the manuscript.
